# Cow dung putrefaction *via* vermicomposting using *Eisenia fetida* and its influence on seed sprouting and vegetative growth of *Viola wittrockiana* (pansy)

Irsa Shafique[1], Saiqa Andleeb[1]*, Farrukh Naeem[2], Shaukat Ali[3], Tauseef Tabassam[4], Tariq Sultan[4], Mohammad Almas Abbasi[5]

**1** Department of Zoology, Microbial Biotechnology and Vermi-Technology Laboratory, Vermi-tech Unit, University of Azad Jammu and Kashmir, Muzaffarabad, Pakistan, **2** First Biotech LLC, Lahore, Pakistan, **3** Department of Zoology, GC University, Lahore, Pakistan, **4** Land Resources Research Institute, Pakistan Agricultural Research Council, Islamabad, Pakistan, **5** Department of Agriculture, Muzaffarabad, Azad Jammu and Kashmir, Pakistan

\* drsaiqa@gmail.com, drsaiqa@ajku.edu.pk

**Data Availability Statement:** All relevant data are within the paper.

**Funding:** Saiqa Andleeb TDF-02-06 Higher Education Commission Pakistan The funders had

## Abstract

The current research was conducted at Vermi-tech Unit, Muzaffarabad in 2018 to evaluate the efficacy of cow dung and vermicompost on seed sprouting, seedlings, and vegetative developmental parameters of *Viola x wittrokiana* (pansy). In the current study, vermicompost was produced using *Eisenia fetida*. Physicochemical parameters of vermicompost and organic manure were recorded before each experimentation. The potting experiment was designed and comprised of eight germination mediums containing different combinations of soil, sand, cow dung, and various concentrations of vermicompost such as 10% VC, 15% VC, 20% VC, 25% VC, 30% VC, and 35% VC. Seed sprouting and seedling developmental parameters were observed for 28 days while vegetative plant growth parameters were recorded after 10 weeks of transplantation in various vermicompost amended germination media. Pre and post-physicochemical analysis of germination media were also recorded to check their quality and permanency. The current findings showed that 30% VC germination media was an effective dose for early seed germination initiation and all seed germination parameters. However, the significant vegetative plant growth and flowering parameters of pansy occurred at 35% VC. Findings revealed that vermicompost not only enhanced the seed germination and growth of pansy but also improved soil health. These results indicate that vermicompost can be exploited as a potent bio-fertilizer for ornamental plant production.

## Introduction

Vermicomposting is a biological and joint process of earthworms and microbes which convert complex organic wastes into a stabilized nutrient-rich fertilizer called vermicompost [1, 2].

no role in study design, data collection and analysis, decision to publish, or preparation of the manuscript.

**Competing interests:** The authors have declared that no competing interests exist.

Vermicomposting (organic farming) is an environmentally friendly, beneficial, and economical technology [3]. Various sources of organic waste i.e. agricultural waste, urban solid waste, animal waste, and agro-industrial waste has been recycled. Vermicomposting is an attractive method for waste management, and is one of the ways to reduce soil pollution and maintain soil fertility [4, 5] Different species of earthworms are used in vermicomposting such as *Eisenia fetida*, *Perionyx excavates*, *Eudrilus eugeniae*, etc [6–10]. In Pakistan, *E. fetida* is the commonly used earthworm species for vermicomposting.

Vermicompost had a significant impact on the growth of various plant species including aromatic plants, medicinal plants, horticultural crops, ornamental plants, cereals, and forestry species [11–18]. The positive outcome of vermicompost on seed germination and seedling parameters was also demonstrated by various researchers [13, 16, 19–21]. According to Ganeshnauth et al. [3], vermicompost is the best organic media containing all nutrients, free of chemicals, free of human and plant pathogens, and can improve the quality of crops, enhance soil fertility and reduce the cost needed to purchase the chemical fertilizers for crop production. Previous literature also demonstrated that vermicompost reduced diseases and pests in crops [22–24]. In the view of above facts, the current experiment was designed to check the impact of cow-dung-based vermicompost on the seedling and vegetative parameters of pansy.

*Viola wittrockian*a (The garden pansy is cultivated in the world as a bedded and pot plant on the largest scale [25]. Modern horticulturists have developed a variety of bi-colors including yellow, gold, orange, white, red, violet, purple, and dark purple. Apart from being used for decorative purposes, both the leaves and flowers are edible and high in vitamins (A and C). The flowers of pansy have been used to make flavored honey, syrup, and salads, and the leaves and flowers can be used as a garnish [26]. They have medicinal importance as well, they are used as anti-allergen, very helpful for cold and flu as it soothes coughs and sore throat, and are also helpful in treating fevers, bronchitis, asthma, and whooping cough [27]. Very long time, pansies have been cultivated in various slow-release peat fertilizers [28]. Startek et al. [29] found that coconut media have a significant impact on the emergence and growth of pansy. Because of inadequate peat resources and great costs of imported germination media, it is desirable to explore for alternating media that are rich in nutrients with good physical properties. Under current conditions, the utilization of vermicompost is a promising organic fertilizer [30, 31]. Thus, vermicompost was used in the current study to screen its impact on seed germination, seedling, and vegetative growth parameters of *Viola wittrockian*a.

## Materials and methods

### Ethical approval

The current study is approved by the ethical committee of Office of Research Innovation and Commercialization (ORIC), the University of Azad Jammu and Kashmir, Muzaffarabad, Pakistan vide no. 99/ORIC/2022: Dated 7/4/2022.

### Waste material and earthworm collection

Animal wastes (cow dung) were collected from local residential houses of Muzaffarabad, Azad Jammu and Kashmir (AJ&K), Pakistan, was air-dried (10 to 15 days) to remove the noxious gases, snails, spiders, and beetles etc.), and sliced into small pieces [32]. The organic waste materials such as grass clippings, rice straw, waste papers, raw fruits and vegetables along with their peels, coconut peel, eggshells, maize straw, and dry leaves were also collected from the local market and residential houses of Muzaffarabad, AJ&K, Pakistan. Pre-digested vegetables and fruits was used for vermicomposting [33]. For vermi-culturing fresh spinach/green vegetables were used. Before making bed for vermicomposting raw papers were dipped in water for

24 h. Locally available earthworms (*E. fetida*) were collected from the Mahajar Camp, a local area of Muzaffarabad, AJ&K, Pakistan, in sealed bags or plastic bottles.

## Experimental design for vermicompost production

The vermicomposting experiment was done in cemented pits (10 x 5 feet) for 90 days. A worm bed was constructed by using moisten papers/grass clippings /shredded twigs, varying according to what is available locally. Feed matter like green leaves, cow dung, wheat straw, debris including remained leaves, crushed eggshells, wasted tea, and tea bags was given to earthworms. The predigested substrate (vegetables and fruits) mixed with the cow dung in a 3:1 ratio. At the beginning of vermicomposting *E. fetida* (100 to 200) were added to the top of the pit. To protect the worm population from predators such as rats, birds, ants, and cockroaches, the pre-compost material was covered by either sieved mesh/banana leaves/cloth cover. Thorough mixing of all materials was done once in a week to confirm the availability of feeding materials for vermi-culturing. The moisture contents and temperature were adjusted *via* sprinkling water. After 90 days, the physical variations such as appearance, texture, color, electrical conductivity, moisture, and temperature were recoded. The earthy smell was observed which indicates the formation of vermicompost. Finally, composting material was passed through a 2 mm sieve and used for measuring the desired parameters.

**Biomass and reproduction (number) of earthworms.** *E. fetida* biomass and the total number of adults, juveniles, and cocoons were noted at the end of vermicomposting. The fecundity rate was also calculated as the number of earthworms/total number of earthworms used x 100.

## Physicochemical analysis of vermicompost and cow dung

Physical parameters of vermicompost and cow dung such as appearance, texture, color, electrical conductivity (EC), moisture, temperature, and the smell were recorded. To accurately measure the chemical parameters affecting composting, standard methods were used for the chemical analysis of each parameter [34]. The total carbon (C%), and nitrogen (N%) were measured using the Walkley-Black method whereas potassium (K%) and phosphorus (P%) levels in the vermicompost and cow dung were measured by the ammonium acetate method and modified Olsen method, respectively [35, 36]. In addition, the concentration of microelements such as magnesium, manganese, sulfates, calcium, copper, iron, and zinc was measured through atomic absorption spectrometry [37, 38].

## Potting experiment

The flexible plastic pots (1 kg capacity) were used to assess the influence of organic manure (cow dung) and prepared vermicompost on seed germination, seedlings, and vegetative growth parameters (shoot length, number of leaves and their measurements, root length etc.) of pansy. The pots were individually filled with the germination medium contain soil (SO), sand (S), cow dung (CD) and various concentrations of cow dung-based vermicompost (VC). In all germination media 0.33 kg sand (S) was used while the details of germination media are given as: To-control (S + 0.67 kg SO), Tcc-control (S + 0.32 kg SO + 0.35 kg CD), 10% VC (S + 0.57 kg SO + 0.1 kg VC), 15% VC (S + 0.52 kg SO + 0.15 kg VC), 20% VC (S + 0.47 kg SO + 0.2 kg VC), 25% VC (S+ 0.42 kg SO + 0.25 kg VC), 30% VC (S + 0.37 kg SO + 0.3 kg VC), and 35% VC (S + 0.32 kg SO + 0.35 kg VC), respectively.

### Pre and post-physicochemical analysis of germination media

To check the stability and quality of germination media before sowing and after harvesting pansy seedlings, physicochemical parameters like color, appearance, texture, electrical conductivity (EC), moisture, temperature, total carbon (%), nitrogen (%), phosphorus (%), and potassium (%) (NPK), were calculated using procedures as declared above.

### Seed sowing and maintenance

Twenty-four pots were prepared and eight pansy seeds were manually sown at a constant distance of 3 cm with equivalent spacing between the seeds in each pot having a 1 kg capacity. Each pot was watered thoroughly each day (morning) so that all the treatments received the same amount of irrigation. The pots were organized randomly in the natural sunlight every week. The radicle emergence was recorded on daily basis. Seeds were handled properly to avoid the damaging effect.

### Observations of seed germination and seedling growth parameters

The seeding day was deliberated as the first day. The efficacy of germination media containing vermicompost on seed germination and seedlings was perceived in the morning each day. The sprouting was calculated each day. Various parameters of seed sprouting i.e. Initiation of seed germination (IoG; First plumule emerged), Mean germination Time (MGT: n1 x d1 + n2 x d2 + n3 x d3 + ———/ Total number of days), Speed of germination (SoG: n1/d1+n2/d2+n3/d3+ ——), Mean daily germination (MDG: Total number of germinated seeds/ Total number of days.), Germination values (GV: PV $_X$ MDG), Peak Value (PV: Highest seed germinated/ Number of days.), Germination percentage (G%: Number of seedlings/total number of seeds x 100), and completion of germination (CoG: the day from which no further germination occurred) were calculated [39–41]. In all parameters "n" represents the number of germinating seeds and "d" number of days. Afterward, 30 days, seedling growth parameters like length of shoot (cm), number of leaves, leaves diameter (cm), leaves length and area (cm), whole seedling length (cm), and length of root (cm), were also recorded.

### Transplantation of pansy plants

The pansy seedlings were transplanted in the pots (7 kg) after 30 days in November 2019 when the temperature (10–15˚C) is superlative for pansy growth. To assess the influence of vermicompost on vegetative growth similar germination media (%) were used as given above and the experiment was conducted in an open area with bright sunlight and active photoperiod of 7–8 h. All plants were harvested at the yield stage and numerous vegetative growth parameters like plant height (cm), root volume, leaves number, leaves diameter (cm), initiation of flowering, number of floral buds, number of open flowers, flower diameter (cm), fresh and dry mass of flowers (g) were investigated. Before transplantation and after harvesting of pansy plants, physicochemical parameters such as electrical conductivity, pH, total organic matter (%), total carbon (%), and (NPK %) were estimated.

### Statistical analysis

All experimentations were done in triplicate. The statistical significance was assessed by one-way analysis of variance (ANOVA) using Graph Pad Prism for Windows (version 5.03). '*' represents a significant change among Tcc and VC-treated groups and '¥' represents a significant change among To and VC-treated groups. Statistical icons: $^*$/¥ = p≤0.05; $^{**}$/¥¥ = p≤0.01; and $^{***}$/¥¥¥ = p≤0.001

## Results

### Physical analysis of vermicompost and cow dung

Vermicompost has been produced on large scale at Vermi-tech Unit, Zoology Department, UAJ&K, Muzaffarabad using various organic waste materials and *E. fetida* earthworm species. After 3 months all the waste material was completely converted into humus-like material with high porosity (vermicompost) *via E. fetida*. Physical properties of cow dung were recorded as pasty and the pile of shaving cream-like appearance, greenish to dark brown color, pungent or rotten egg-like smell, 18% moisture contents, 8.7 pH, 55–60˚C temperature, and 3.52 mS/cm EC, respectively. On the other hand, the physical properties of prepared vermicompost were recorded as fine grain and porous appearance with an earthy smell, blackish-brown in color, 7.8 pH, 25˚C temperature, 2.82 mS/cm EC, and 25% moisture contents, respectively.

### Chemical analysis of vermicompost and cow dung

The current research revealed that vermicompost possessed a significantly ($p \leq 0.001$) high level of nutrients such as NPK, calcium, and magnesium compared to cow dung manure. The total organic carbon and organic matter were significantly ($p \leq 0.001$) reduced (25.2% and 43%) in vermicompost compared to cow dung (50.4% and 86%). On the other hand, the percentage of NPK was increased in vermicompost (2.28%, 0.030%, and 0.033%) compared to cow dung (1.86%, 0.0098%, and 0.020), respectively. The cation exchange capacity of vermicompost was greater (72.48 me/100g) compared to cow dung, 62.30 me/100g. The carbon to nitrogen ratio (C: N) was reduced in vermicompost (11:1) paralleled to cow dung (26:1), respectively. The level of microelements such as sulfates (0.55%), zinc (0.110%), iron (1.3313%), manganese (0.2038%), copper (0.0048%), magnesium (0.78%), and calcium (1.35%) were also increased in vermicompost compared to the cow dung such as sulfates (0.35%), zinc (0.0012%), iron (1.1690%), manganese (0.0414%), copper (0.0017%), magnesium (0.25%), and calcium (1.0%).

### Reproduction and growth of *E. fetida*

The growth and reproduction of *E. fetida* were recorded at the beginning of the experiment, after 45 days, and after 90 days (Fig 1). The maximum adult earthworms were recorded at the end of vermicomposting. Initially, the biomass of used earthworms was recorded as 15 mg but at the end of experimentation the biomass was recorded as 300 mg. Juveniles were not seen in the first week of vermicomposting but after 90 days the maximum number of juveniles were recorded. Cocoon production was increased due to favorable environmental conditions. The current study reveals that the maximum fecundity was recorded during vermicomposting.

### Physicochemical analysis of soil and sand before sowing

Results revealed that the soil used for marigold seed germination before sowing was sandy loam in texture, the brown color having pH 7.64, and electrical conductivity 2.0 mS/cm. Chemically the soil consisted of 0.0014% potassium, 0.61% organic carbon, 1.049% organic matter, 0.52% nitrogen and 0.0072% phosphorus. Similarly, the chemical analysis of sand used for the experiment showed 1.53% organic carbon, 2.64% organic matter, 0.001% phosphorus,1.32% nitrogen, and 0.0112% potassium.

### Physicochemical analysis of germination media before sowing

Results revealed that the addition of vermicompost influences both physical and chemical properties of germinating media compared to controls (To: soil + sand and Tcc: soil + sand

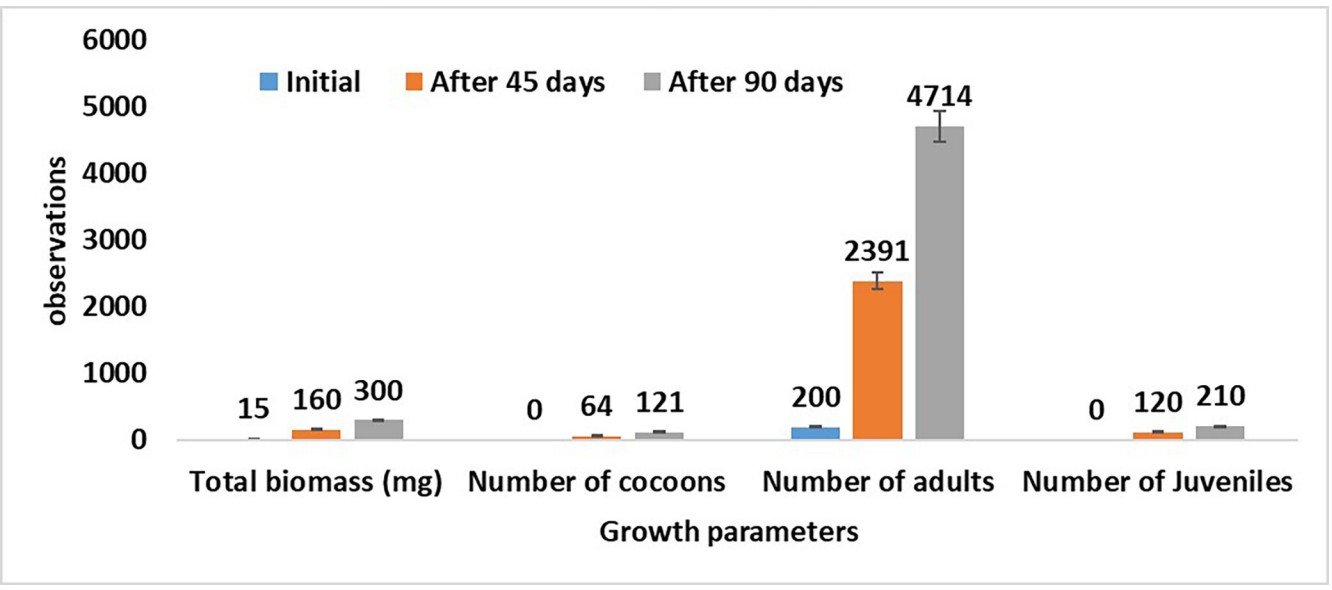

**Fig 1. Influence of waste products on *E. fetida* fecundity and growth of during vermicomposting.**

+ cow dung) (Table 1). It was observed that the pH of germination media was slightly decreased (7.6±0.0 to 7.4±0.0) in all treatments after the addition of vermicompost. However, the pH value in control cow dung (Tcc) was 8.7±0.0. Results revealed that Tcc media was crumpled, thick, and greenish-brown, while To germinating media was hard, compact, less porous, and blackish-brown. On the other hand, all VC-containing germinating media was blackish-brown with high porosity and soft aerated texture. Electrical conductivity was recorded as 2.20 ±0.05 mS/cm and 3.48 ±0.01 mS/cm in To and Tcc while slightly reduced in all vermicompost containing germination media (from 2.80±0.02mS/cm to 2.73±0.03 mS/cm), respectively. The average percentage of carbon and nitrogen was increased in VC-containing germination media compared to controls. Similarly, average phosphorus noted in 35% was 0.062±0.00% followed by 30% (0.051±0.00%) and 25% (0.051±0.00%). The average potassium contents in 35% were recorded as 0.47±0.05%. On the other hand, 10%, 15%, 20%, 25%, and 30% showed average potassium contents as 0.024±0.00%, 0.051±0.00%, 0.031±0.00%, 0.054±0.00% and 0.533 ±0.00% respectively.

## Impact of vermicompost on seed germination

Daily observations were made (for 28 days) on the emergence of radicles to analyze the impact of vermicompost on pansy seed germination. Considerable and significant ($p \leq 0.05$ and $p \leq 0.001$) variations among germination parameters at different concentrations of vermicompost were calculated (Table 2). Results revealed that early initiation of seed germination was recorded in 30% VC concentration (3.0±0.0 days) compared to other germination media (Table 2). The maximum mean generation time was recorded in Tcc and To (8.56±0.39 days and 8.10±0.02 days) while the minimum mean generation time was recorded in 30% VC (3.63 ±0.012 days). The maximum mean daily germination of seeds was found in 30% VC (0.32 ±0.02 days) and 25% VC (0.30±0.05 days) while the minimum value was found in To (0.21 ±0.01 days) and Tcc (0.22±0.02 days). The speed of germination was highest at 30% VC (9.63 ±0.08) and 35% VC (9.31±0.01). The lowest speed of germination was observed in Tcc (5.33 ±0.08) and To (4.81±0.08). Data revealed that the minimum days were taken for completion of

**Table 1. Pre and post physicochemical examination of germinating media used for pansy seed germination and seedling.**

| Parameters→ / Treatments↓ | | Physicochemical analysis | pH | Appearance | Electrical conductivity (mS/cm) | Carbon (%) | Nitrogen (%) | Patassium (%) | Phosphorous (%) |
|---|---|---|---|---|---|---|---|---|---|
| Controls | Tcc | PS-A | 8.7±0.0 | TC, G | 3.48 ±0.01 | 0.80±0.05 | 0.83±0.05 | 0.01±0.00 | 0.038±0.00 |
| | | PH-A | 8.7±0.0 | TC, GB | 3.38±0.10 | 0.61±0.01 | 0.66±0.01 | 0.01±0.00 | 0.029±0.00 |
| | To | PS-A | 7.6±0.0 | BB, C, LP | 2.20±0.05 | 0.97±0.01 | 0.83±0.05 | 0.01±0.00 | 0.038±0.00 |
| | | PH-A | 7.6±0.0 | BB, C, LP | 2.17±0.37 | 0.82±0.01 | 0.66±0.05 | 0.01±0.00 | 0.018±0.00 |
| Vermicompost | 10% | PS-A | 7.4±0.0[***,¥¥¥] | BB, P | 2.80 ±0.02[***,¥¥¥] | 1.30±0.05[***,¥¥¥] | 1.01±0.01[***,¥¥¥] | 0.02±0.00[***,¥¥] | 0.041±0.00[***,¥¥¥] |
| | | PH-A | 7.4±0.0[***,¥¥¥] | BB, P | 2.71±0.01[***,¥¥¥] | 1.14±0.01[***,¥¥¥] | 0.90±0.01[***,¥¥¥] | 0.02±0.00[***,¥¥] | 0.024±0.00[aa,¥¥¥] |
| | 15% | PS-A | 7.4±0.0[***,¥¥¥] | BB, P | 2.77±0.02[***,¥¥¥] | 1.34±0.01[***,¥¥¥] | 1.13±0.05[***,¥¥¥] | 0.051±0.00[***,¥¥¥] | 0.044±0.00[***,¥¥¥] |
| | | PH-A | 7.4±0.0[***,¥¥¥] | BB, P | 2.72±0.02[***,¥¥¥] | 1.21±0.01[***,¥¥¥] | 0.94±0.01[***,¥¥¥] | 0.040±0.00[***,¥¥¥] | 0.035±0.00[***,¥¥¥] |
| | 20% | PS-A | 7.4±0.0[***,¥¥¥] | BB, P | 2.73±0.03[***,¥¥¥] | 1.44±0.01[***,¥¥¥] | 0.57±0.05[***,¥¥¥] | 0.03±0.00[***,¥¥] | 0.026±0.00[***,¥¥¥] |
| | | PH-A | 7.4±0.0[***,¥¥¥] | BB, P | 2.59 ±0.01[***,¥¥¥] | 1.24±0.05[***,¥¥¥] | 0.33±0.01[***,¥¥¥] | 0.022±0.00[***,¥¥¥] | 0.018±0.00[***,¥¥¥] |
| | 25% | PS-A | 7.4±0.0[***,¥¥¥] | BB, P | 2.73±0.03[***,¥¥¥] | 1.40±0.05[***,¥¥¥] | 1.01±0.01[***,¥¥¥] | 0.051±0.00[***,¥¥¥] | 0.054±0.00[***,¥¥¥] |
| | | PH-A | 7.4±0.0[***,¥¥¥] | BB, P | 2.54±0.05[***,¥¥¥] | 1.18±0.01[***,¥¥¥] | 0.89±0.01[***,¥¥¥] | 0.04±0.00[***,¥¥] | 0.031±0.00[***,¥¥¥] |
| | 30% | PS-A | 7.4±0.0[***,¥¥¥] | BB, P | 2.73±0.03[***,¥¥¥] | 1.51±0.01[***,¥¥¥] | 1.23±0.05[***,¥¥¥] | 0.533±0.00[***,¥¥¥] | 0.051±0.00[***,¥¥¥] |
| | | PH-A | 7.4±0.0[***,¥¥¥] | BB, P | 2.49±0.05[***,¥¥¥] | 1.32±0.01[***,¥¥¥] | 0.3±0.01[***,¥¥] | 0.01±0.00[***,¥¥] | 0.014±0.00[***,¥¥¥] |
| | 35% | PS-A | 7.4±0.0[***,¥¥¥] | BB, P | 2.73±0.03[***,¥¥¥] | 1.35±0.05[***,¥¥¥] | 0.94±0.01[***,¥¥¥] | 0. 47±0.05[***,¥¥¥] | 0.062±0.00[***,¥¥¥] |
| | | PH-A | 7.4±0.0[***,¥¥¥] | BB, P | 2.49±0.05[***,¥¥¥] | 1.17±0.02[***,¥¥¥] | 0.53±0.01[***,¥¥¥] | 0.030±0.00[***,¥¥¥] | 0.034±0.00[***,¥¥¥] |

Pre sowing analysis (**PS-A**), Post harvesting analysis (**PH-A**), Thick crumpled (**TC**), greenish brown (**GB**), Blackish brown (**BB**), compact (**C**), less porous, (**LP**), porous (**P**), Statistical icons

[***]/[¥¥¥] = p≤0.001

**Table 2. Seed germination parameters in pansy under different compositions of germination media.**

| Germination parameters → / Treatment ↓ | | IoG (Days) | MGT | MDG | SoG | CoG (Days) | (G%) | PV | GV |
|---|---|---|---|---|---|---|---|---|---|
| Controls | Tcc | 6.66±0.47 | 8.56±0.39 | 0.22±0.02 | 5.33±0.08 | 8.00±0.0 | 75.6±0.47 | 0.93±0.01 | 0.237±0.0 |
| | T0 | 6.66±0.47 | 8.10±0.02 | 0.21±0.01 | 4.81±0.08 | 8.66±0.47 | 63.0±0.47 | 0.52±0.01 | 0.11±0.05 |
| Vermicompost | 10% | 4.33±0.47[***,¥¥¥] | 7.54±0.03[***,¥] | 0.25±0.01[*,¥] | 7.83±0.09[***,¥¥¥] | 8.33±0.47 | 81.5±1.22[***,¥¥¥] | 0.83±0.01[***,¥¥¥] | 0.21±0.12[***,¥¥¥] |
| | 15% | 4.33±0.47[***,¥¥¥] | 7.02±0.02[***,¥¥¥] | 0.24±0.02[*,¥] | 8.10±0.0[***,¥¥¥] | 7.66±0.47 | 81.6±0.94[***,¥¥¥] | 0.93±0.01[***,¥¥¥] | 0.263±0.05[***,¥¥¥] |
| | 20% | 4.00±0.0[***,¥¥¥] | 6.50±0.05[***,¥¥¥] | 0.31±0.02[***,¥¥¥] | 8.48±0.01[***,¥¥¥] | 7.00±0.0[***,¥¥¥] | 91.6±1.24[***,¥¥¥] | 1.33±0.0[***,¥¥¥] | 0.40±0.05[***,¥¥¥] |
| | 25% | 3.33±0.47[***,¥¥¥] | 4.22±0.0[***,¥¥¥] | 0.30±0.05[***,¥¥¥] | 9.12±0.02[***,¥¥¥] | 8.33±0.47 | 96.3±0.47[***,¥¥¥] | 1.02±0.01[***,¥¥¥] | 0.31±0.05[***,¥¥¥] |
| | 30% | 3.00±0.0[***,¥¥¥] | 3.63±0.01[***,¥¥¥] | 0.32±0.02[***,¥¥¥] | 9.63±0.08[***,¥¥¥] | 6.33±0.47 | 100±0.0[***,¥¥¥] | 1.64±0.01[***,¥¥¥] | 0.52±0.01[***,¥¥¥] |
| | 35% | 4.33±0.47[***,¥¥¥] | 4.16±0.01[***,¥¥¥] | 0.25±0.00[*,¥] | 9.31±0.01[***,¥¥¥] | 8.33±0.47 | 76.0±0.0[***,¥¥¥] | 0.85±0.00[***,¥¥¥] | 0.212±0.0[***,¥¥¥] |

Initiation of germination (**IoG**), Speed of germination (**SoG**), Mean germination time (**MGT**), Mean daily germination (**MDG**), Germination value (**GV**), Completion of germination (**CoG**), Germination percentage (**G%**), Peak value (**PV**), Statistical icons: [*]/[¥] = p≤0.05 and [***]/[¥¥¥] = p≤0.001

germination in 30% VC (6.33±0.47 days) compared to other germination media compositions (Table 2). The maximum percentage of germination (100.0±0.0%) was recorded at 30% VC followed by (96.3±0.47%) at 25% VC > (91.6±1.24%) at 20% VC > (81.6 ±0.94%) at 15% VC > (81.5 ±1.22%) at 10% VC > (76.0±0.0%) at 35% VC > (75.6±0.47%) at Tcc, respectively (Table 2). However, the minimum germination percentage was recorded in To (63.0±0.0%). The maximum peak value was observed in 30% VC (1.64±0.01) compared to other germination media. The maximum germination value was recorded at 30% VC (0.52±0.01) followed by 20% VC (0.40±0.05) while the minimum germination value was recorded in To (0.11 ±0.05), in Tcc (0.237±0.0), and at 10% VC (0.21±0.12). The whole results of seed germination indicate that all germination parameters were maximum at 30% VC germination media compared to others. Thus, the current research indicates that 30%VC is the preferable germination media for pansy seeds. The impact of various vermicompost containing germination media and controls (To and Tcc) on pansy seed germination is shown in Fig 2.

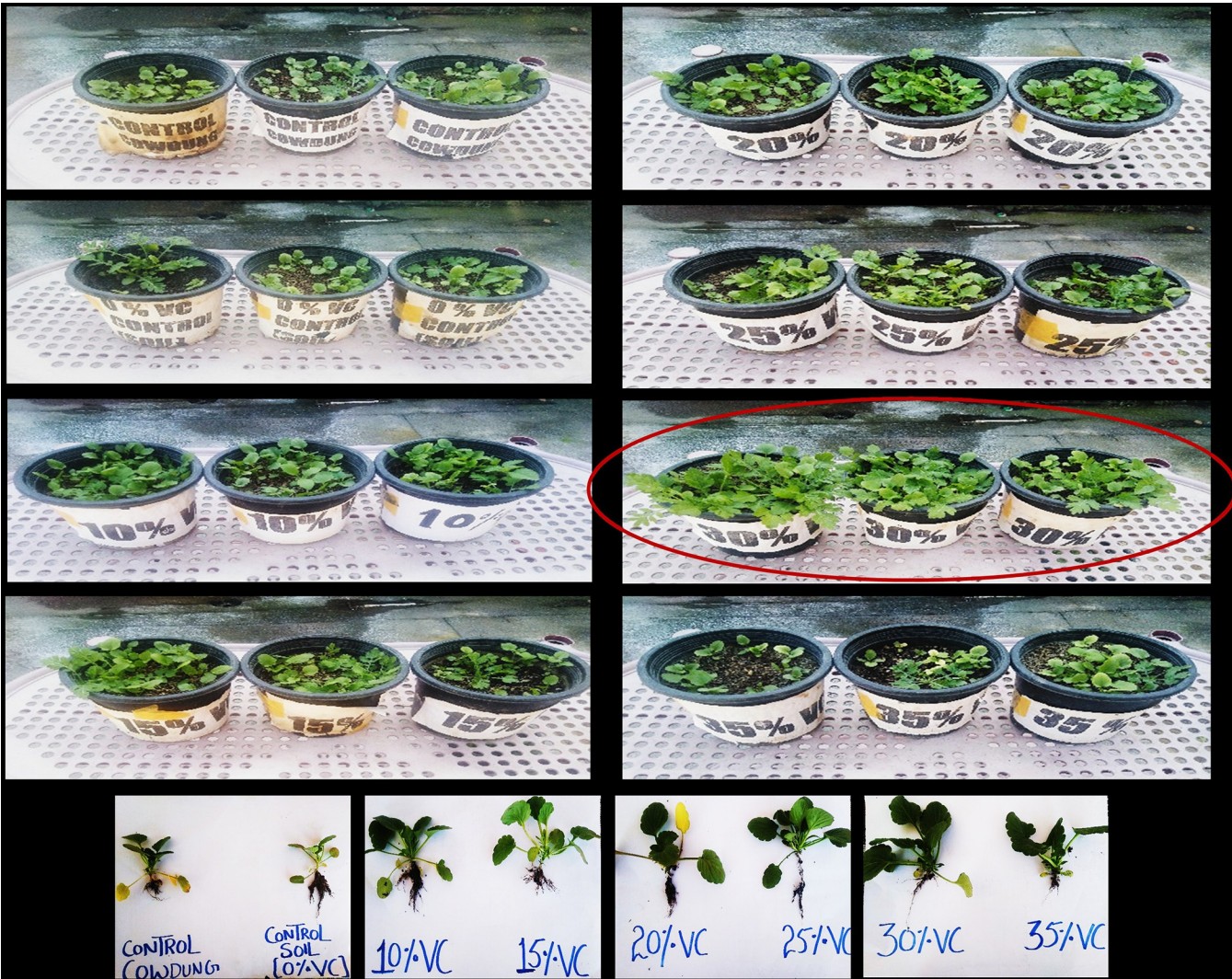

**Fig 2. Influence of vermicompost based germinating media on pansy seeding.**

## Seedling growth parameters

A significant (p≤0.001) increase in all growth parameters was recorded in VC-treated plants when compared to control plants (Fig 2). Results revealed that 30% VC-containing germinating media showed a significant increase in the number of leaves (7.33±0.47), shoot length (3.66±0.47 cm), the diameter of leaves (3.00 ±0.0 cm), length of leaves (3.66±0.47 cm), leaf area (6.33±0.47 cm), whole seedling length (9.00±0.0 cm), and root length (4.66±0.47 cm) of pansy (Fig 2; Table 3). Overall, it was observed that vermicompost has a significant impact on the pansy seedlings compared to controls (Table 3; Fig 2).

## Physicochemical analysis of germination media after seedling harvesting

After harvesting pansy seedlings, the germinating media was again analyzed to check the remaining nutrient profile. All germination media showed significant (p≤0.001) decreased values of phosphorus, nitrogen, carbon, and potassium compared to prepared media used for seed sowing (Table 1). However, pH values such as 8.7±0.0 in Tcc, 7.6±0.0 in To, and 7.4±0.0 in all VC-treatments remained the same. Electrical conductivity was slightly reduced in To and Tcc (2.17±0.37 mS/cm and 3.38±0.10 mS/cm) respectively. Similarly, EC was also reduced in all VC-treated (2.71±0.01 mS/cm to 2.49±0.05 mS/cm) at p≤0.001.

## Physicochemical properties of the germinating media before transplantation of seedling

The pH, EC, and nutrient contents (N, P, K, C) in germinating media were analyzed before the transplantation of the seedling (Table 4). The pH was recorded in Tcc and To as 8.7±0.0 and 7.6±0.0 whereas 7.4±0.0 was recorded in all VC-treated. The appearance and texture vary in controls and vermicompost treatments, i.e. Tcc was crumpled with greenish-brown color and thick, To was compact, less porous having brownish-black color whereas all vermicompost treatments were soft aerated and blackish-brown with high porosity, respectively. Electrical conductivity in To and Tcc was recorded as 2.13±0.01 mS/cm and 3.45±0.05 mS/cm while a range between 2.84±0.01 mS/cm to 2.97±0.02 mS/cm was recorded in all VC-treated.

## Vegetative growth parameters

The vegetative growth parameters (plant height, volume of root, number of shoots / plant, and length of roots and shoots) and floral parameters (No. of flowers, no. of floral buds, the

**Table 3. Pansy seedling growth parameters.**

| Seedling parameters → | | LoS (cm) | NoL/P | DoL (cm) | LoL (cm) | LA (L x B x K) (cm) | WSL (cm) | RL (cm) |
|---|---|---|---|---|---|---|---|---|
| **Treatment ↓** | | | | | | | | |
| **Controls** | **T₀** | 1.66±0.47 | 4.33±0.47 | 2.0 ±0.0 | 1.16±0.23 | 1.53±0.37 | 3.66±0.47 | 1.66±0.47 |
| | **Tcc** | 1.66±0.47 | 5.33±0.47 | 2.0±0.0 | 1.66±0.47 | 2.26±0.18 | 5.33±0.47 | 2.66 ±0.47 |
| **Vermicompost** | **10%** | 2.33±0.47 | 4.66±0.47 | 2.0±0.0 | 2.16±0.23 | 2.5±0.40 | 4.66±0.47 | 2.66±0.47 |
| | **15%** | 2.33±0.47 | 5.66±0.47 | 2.0±0.0 | 2.33±0.47 | 2.5±0.75 | 4.33±0.47 | 2.66±0.47 |
| | **20%** | 2.66±0.47 | 6.0±0.0 | 2.16±0.23 | 2.33±0.47 | 3.53±0.40 | 4.33±0.47 | 2.66±0.47 |
| | **25%** | 3.33±0.47 | 6.66±0.47 | 2.16 ±0.23 | 3.5±0. 40***,¥¥¥ | 4.0±0.0 | 7.00±0.0***,¥¥¥ | 3.66±0.47***,¥¥¥ |
| | **30%** | 3.66±0.47***,¥¥¥ | 7.33±0.47***,¥¥¥ | 3.0 ±0.0***,¥¥¥ | 3.66±0.47***,¥¥¥ | 6.33±0.47***,¥¥¥ | 9.00±0.0***,¥¥¥ | 4.66±0.47***,¥¥¥ |
| | **35%** | 2.66±0.47 | 6.33±0.47 | 1.83±0.23 | 2.33±0.47 | 3.83±0.23 | 6.33±0.47 | 3.00±0.0 |

Root length (**RL**), Length of shoot (**LoS**), Diameter of leaves (**DoL**), Total number of leaves / plant (**NoL/P**), Length of leaves (**LoL**), Leaf area (**LA**), Whole Seedling length (**WSL**), Statistical icons: ***/¥¥¥ = p≤0.001

**Table 4. Pre and post physicochemical examination of germinating media used for pansy vegetative growth parameters.**

| Parameters→ Treatments ↓ | | Physicochemical analysis | pH | Appearance | Electrical conductivity (mS/cm) | Carbon (%) | Nitrogen (%) | Patassium (%) | Phosphorous (%) |
|---|---|---|---|---|---|---|---|---|---|
| **Controls** | **Tcc** | **PT-A** | 8.7±0.0 | TC, GB | 3.45±0.05 | 0.83±0.23 | 0.10±0.05 | 0.0071±0.00 | 0.040±0.00 |
| | | **PH-A** | 8.7±0.0 | TC, GB | 3.34±0.02 | 0.5±0.08 | 0.07±0.05 | 0.0065±0.00 | 0.034±0.00 |
| | **To** | **PT-A** | 7.6±0.0 | BB, C, LP | 2.13±0.01 | 0.86±0.05 | 0.10±0.05 | 0.0074±0.00 | 0.040±0.00 |
| | | **PH-A** | 7.6±0.0 | BB, C, LP | 2.07±0.09 | 0.5±0.05 | 0.06±0.05 | 0.0062±0.00 | 0.032±0.00 |
| **Vermicompost** | **10%** | **PT-A** | 7.4 ±0.0[***],[¥¥¥] | BB, P | 2.84±0.01[***],[¥¥¥] | 0.73±0.05[*],[¥] | 0.066 ±0.00[***],[¥¥¥] | 0.021 ±0.00[***],[¥¥¥] | 0.039±0.00[***],[¥¥] [¥] |
| | | **PH-A** | 7.4 ±0.0[***],[¥¥¥] | BB, P | 2.74±0.01[***],[¥¥¥] | 0.46 ±0.05[***],[¥] | 0.052 ±0.00[***],[¥¥¥] | 0.021 ±0.00[***],[¥¥¥] | 0.031±0.00[***],[¥¥] [¥] |
| | **15%** | **PT-A** | 7.4 ±0.0[***],[¥¥¥] | BB, P | 2.80±0.00[***],[¥¥¥] | 0.73±0.05[a,b] | 0.066 ±0.00[***],[¥¥¥] | 0.044 ±0.00[***],[¥¥¥] | 0.036±0.00[***],[¥¥] [¥] |
| | | **PH-A** | 7.4 ±0.0[***],[¥¥¥] | BB, P | 2.70±0.05[***],[¥¥¥] | 0.6±0.08[***],[¥] | 0.055 ±0.00[***],[¥¥¥] | 0.035 ±0.00[***],[¥¥¥] | 0.028±0.00[***],[¥¥] [¥] |
| | **20%** | **PT-A** | 7.4 ±0.0[***],[¥¥¥] | BB, P | 2.84±0.02[***],[¥¥¥] | 1.23 ±0.05[***],[¥¥¥] | 0.10 ±0.00[***],[¥¥¥] | 0.051 ±0.00[***],[¥¥¥] | 0.043±0.00[***],[¥¥] [¥] |
| | | **PH-A** | 7.4 ±0.0[***],[¥¥¥] | BB, P | 2.74±0.01[***],[¥¥¥] | 0.83 ±0.05[***],[¥] | 0.07 ±0.01[***],[¥¥¥] | 0.042 ±0.00[***],[¥¥¥] | 0.032±0.00[***],[¥¥] [¥] |
| | **25%** | **PT-A** | 7.4 ±0.0[***],[¥¥¥] | BB, P | 2.90±0.01[***],[¥¥¥] | 1.01 ±0.01[***],[¥¥¥] | 0.10 ±0.00[***],[¥¥¥] | 0.05±0.00[***],[¥] [¥¥] | 0.049±0.00[***],[¥¥] [¥] |
| | | **PH-A** | 7.4 ±0.0[***],[¥¥¥] | BB, P | 2.80±0.05[***],[¥¥¥] | 0.63 ±0.05[***],[¥] | 0.08 ±0.05[***],[¥¥¥] | 0.041 ±0.00[***],[¥¥¥] | 0.042±0.00[***],[¥¥] [¥] |
| | **30%** | **PT-A** | 7.4 ±0.0[***],[¥¥¥] | BB, P | 2.92±0.05[***],[¥¥¥] | 0.76±0.05[**],[¥¥] | 0.066 ±0.00[***],[¥¥¥] | 0.035 ±0.00[***],[¥¥¥] | 0.059±0.00[***],[¥¥] [¥] |
| | | **PH-A** | 7.4 ±0.0[***],[¥¥¥] | BB, P | 2.84±0.01[***],[¥¥¥] | 0.53 ±0.05[***],[¥] | 0.052 ±0.00[***],[¥¥¥] | 0.055 ±0.00[***],[¥¥¥] | 0.026±0.00[***],[¥¥] [¥] |
| | **35%** | **PT-A** | 7.4 ±0.0[***],[¥¥¥] | BB, P | 2.97±0.02[***],[¥¥¥] | 0.366±0.05[*],[¥] | 0.033 ±0.00[***],[¥¥¥] | 0.046 ±0.00[***],[¥¥¥] | 0.030±0.00[aa, ¥¥] |
| | | **PH-A** | 7.4 ±0.0[***],[¥¥¥] | BB, P | 2.86±0.05[***],[¥¥¥] | 0.23 ±0.05[***],[¥] | 0.05 ±0.05[***],[¥¥¥] | 0.028 ±0.00[***],[¥¥¥] | 0.021±0.00[***],[¥¥] [¥] |

Pre transplantation analysis (**PT-A**), Post harvesting analysis (**PH-A**), Thick crumpled (**TC**), greenish brown (**GB**), Blackish brown (**BB**), compact (**C**), less porous (**LP**), porous (**P**), **Statistical icons:** [*]/¥ = p≤0.05; [**]/¥¥ = p≤0.01; and [***]/¥¥¥ = p≤0.001

diameter of flower, fresh and dry weight of flowers) showed significant effects (p≤0.001) in case of vermicompost containing germination media compared to controls Tcc and To (Fig 3; Table 5). The maximum plant height (12.33±0.47 cm) was observed in 35% VC germinating media while the minimum height of the plant was obtained in Tcc (3.5±0.40 cm) and To (3.16 ±0.23 cm). The maximum root volume 7.33±0.47 and 6.66±0.47 were recorded at 35% VC and 25% VC while the minimum root volume was recorded in To (3.66±0.47). The maximum number of leaves / plant was observed at 35% VC (22.3±0.47) and the minimum number of leaves / plant was observed in Tcc (19.00 ±0.81) and To (18.33 ±1.27). Table 5 reveals the minimum average diameter of pansy plants grown in To and Tcc (2.16±0.23 cm and 2.33±0.23 cm), while the maximum average diameter of leaves was recorded in 35% VC (4.16±0.23) cm. A maximum number of open flowers / plant was recorded at 35% VC (3.33±0.47) whereas the minimum number of flowers was recorded in Tcc (1.00±0.0) and To (1.66±0.47). The maximum number of floral buds was observed in 35% VC (3.33±0.22) and the minimum number of floral buds was observed in control Tcc (1.00±0.0). The maximum diameter (7.83±0.23 cm) of a pansy flower was observed at 35% VC whereas a minimum diameter (3.66 ±0.47) cm was observed in control Tcc. The maximum fresh weight (13.66±0.47 g) of flowers was observed in 35% VC whereas control To and control Tcc showed a minimum fresh weight of flowers i.e.

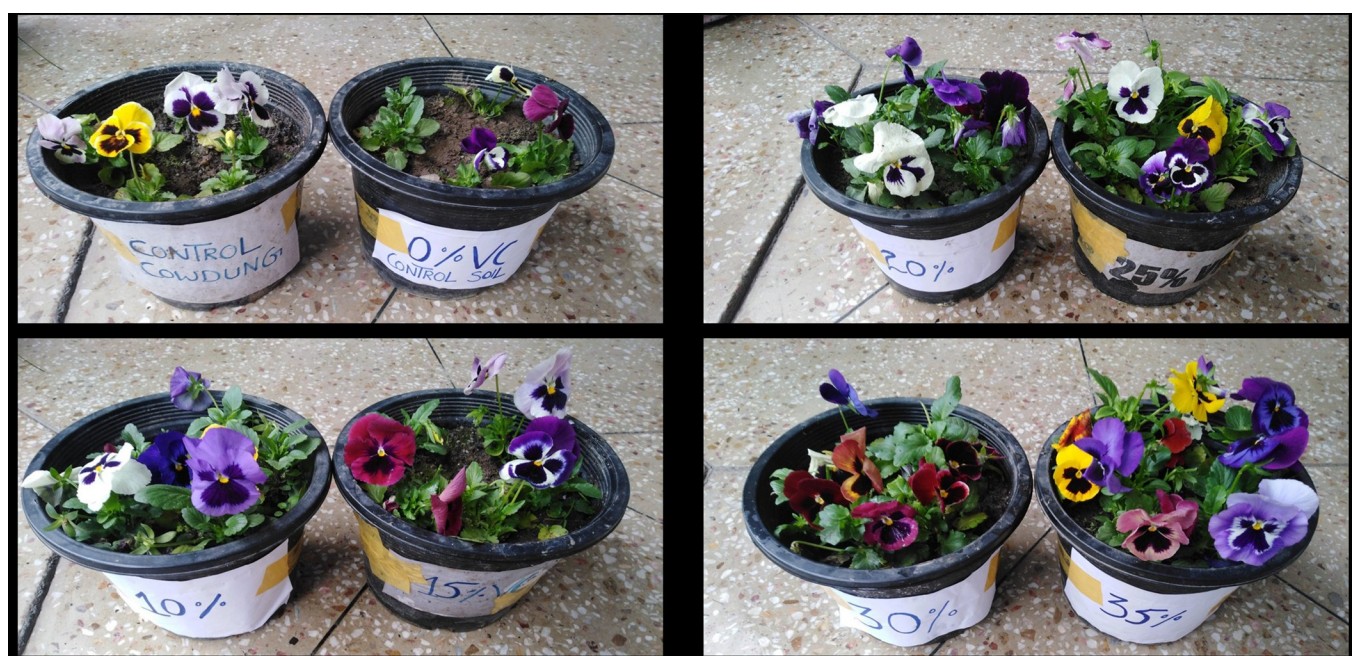

**Fig 3. Influence of vermicompost based germinating media on vegetative growth of pansy at 15<sup>th</sup> week of transplantation.**

2.33±0.23 g and 1.83±0.23 g. The maximum dry weight (7.2±0.08) of flowers was observed in 35% VC whereas To and Tcc showed the minimum dry weight of flowers i.e. 0.3±0.00 g and 0.1±0.00 g respectively. From the whole results, we can say that for the development of plant growth parameters 35% VC germinating media is the best media as shown in Fig 3.

**Table 5. Effect of germination media on vegetative plant growth and flowering parameters of pansy.**

| Growth and flowering parameters → Treatment ↓ | | PH (cm) | VoR (cm³) | NoL/P | DoL (cm) | NoF/P | NoFB | DoF (cm) | FWoF/P (g) | DWoF/P (g) |
|---|---|---|---|---|---|---|---|---|---|---|
| **Controls** | Tcc | 3.5±0.40 | 4.16±0.62 | 19.00±0.81 | 2.33±0.23 | 1.00±0.0 | 1.00±0.0 | 3.66±0.47 | 1.83±0.23 | 0.1±0.0 |
| | T₀ | 3.16±0.23 | 3.66±0.47 | 18.33±1.27 | 2.16±0.23 | 1.66±0.47 | 1.66±0.47 | 6.8±0.05 | 2.33±0.23 | 0.3±0.0 |
| **Vermicompost** | 10% | 11.00±0.8***,¥¥¥ | 5.33±0.47***,¥¥¥ | 21.00±0.81***,¥¥¥ | 3.88±0.23ᵃᵃ,ᵇᵇ | 2.00±0.0 | 2.33±0.47 | 6.8±0.23*** | 6.33±0.47***,¥¥¥ | 3.73±0.09***,¥¥¥ |
| | 15% | 6.66±0.47***,¥¥¥ | 4.16±0.23 | 14.00±0.81 | 2.83±0.25 | 2.33±0.47 | 2.66±0.94 | 5.66±0.47*** | 8.5±0.40***,¥¥¥ | 4.83±0.12***,¥¥¥ |
| | 20% | 10.66±0.47***,¥¥¥ | 5.5±0.40***,¥¥¥ | 13.3±0.47 | 4.00±0.0***,¥¥¥ | 3.00±0.0***,¥¥¥ | 3.00±0.0***,¥¥¥ | 6.16±0.23*** | 11.6±0.47***,¥¥¥ | 6.00±0.0***,¥¥¥ |
| | 25% | 9.5±0.40***,¥¥¥,¥ | 6.66±0.47***,¥¥¥ | 21.33±1.24***,¥¥¥ | 3.33±0.47 | 2.66±0.47 | 3.00±0.0***,¥¥¥ | 6.5±0.40*** | 10.83±0.23***,¥¥¥ | 5.16±0.12***,¥¥¥ |
| | 30% | 10.66±0.47***,¥¥¥ | 4.66±0.47 | 12.33±0.47 | 2.5±0.40 | 2.66±0.47 | 2.33±0.47 | 6.66±0.47*** | 10.83±0.62***,¥¥¥ | 5.16±0.23***,¥¥¥ |
| | 35% | 12.33±0.47***,¥¥¥ | 7.33±0.47***,¥¥¥ | 22.3±0.47***,¥¥¥ | 4.16±0.23***,¥¥¥ | 3.33±0.47***,¥¥¥ | 3.33±0.47***,¥¥¥ | 7.83±0.23***,¥¥¥ | 13.66±0.47***,¥¥¥ | 7.2±0.08***,¥¥¥ |

Plant height (**PH**), Volume of root (**VoR**), No. of leaves / plant (**NoL/P**), Diameter of leaves (**DoL**), Number of open flowers / plant (**NoF/P**), Number of floral buds (**NoFB**), Diameter of flower (**DoF**), Fresh weight of flower /plant (**FWoF/P**), Dry weight of flower /plant (**DWoF/P**), Statistical icons: ***/¥¥¥ = p≤0.001

### Post harvesting physicochemical analysis of the germinating media

The nutrient contents (N, P, K, C) were again analyzed after harvesting pansy plants to check the nutrient efficiency of germinating media. The pH value of Tcc and To (8.7±0.0 and 7.6 ±0.0) was recorded. Similarly, the same pH (7.4±0.0) were recorded in all VC-treated. The appearance and texture of Tcc were crumpled, greenish-brown color, and thick, To was less porous, compact, and hard with brownish-black color, while ermicompost based germinating media were soft aerated, highly porous, and blackish-brown in color. Electrical conductivity in To and Tcc was some what reduced (from 2.13±0.01 mS/cm to 2.07±0.09 mS/cm and from 3.45±0.05 mS/cm to 3.34±0.02 mS/cm). Similarly, electrical conductivity in all vermicomposting media was also declined and recorded in a range 2.74±0.01 mS/cm to 2.86±0.05 mS/cm. Germinating media having vermicompost showed a significant decline in C, N, P, and K when matched to the nutrient contents of before transplantation germinating media (Table 4).

## Discussion

The findings of the current study showed that the seedling and growth parameters were significantly increased when treated with vermicompost compared to the control treatments i.e cow dung and soil. These observations are following the previous reports [15, 16, 42]. In our study vermicompost was prepared using cow dung under similar non-thermophilic moisture-loving conditions *via* utilizing *Eisenia fetida*. Enough space for vertical and horizontal movements was provided for earthworms and our resultant product was blackish-brown, lightweight, and a porous structure having a soft granular appearance. current findings agreed with the outcomes of Acıkbas and Belliturk [43]. Swati and Hait [44] illustrated that vermicompost is cost-effective, an eco-friendly, and perfect soil improvement fertilizer. Numerous literature demonstrated that the earthworms' utilization increases soil organic matter [45, 46]. Dominguez et al. [47] described that earthworms enhance especially in soil quality, microbial activity and crop production. In the current research, *E. fetida* has been used for vermicompost production. According to the Belliturk [48] and Belliturk et al. [49], *E. fetida* is appropriate for outdoor vermicomposting.

Vermicompost stimulates plant growth and improved yield due to high water holding capacity, maximum nutrient (NPK) concentrations, and microbial actions. Our outcomes agreed with Soobhany et al. [50] and Zhu et al. [51]. They showed that vermicompost amended soil has remarkable biological and physicochemical parameters such as more porosity, aeration, nutrients, and organic matters, considerable pH, temperature, moisture contents, and conductivity. At the end of the experiment, the physical parameters were recorded as temperature25˚C, 2.82 mS/cm EC, 7.8 pH, and 25% moisture, respectively. The outcomes of the current research agreed with the findings of Khater [52], who suggested that different compost have various ranges of moisture contents from 23.5% to 32.10% and 2.82mS/cm EC. Shak et al. [53] illustrated the decrease in EC during vermicomposting, and this may be due to the mineralization of organic acids and leaching or precipitation of soluble salts. On the other hand, physical analysis of cow dung showed 30–45˚C temperature, 3.52 mS/cm EC, 8.7pH, and 18% moisture contents which are in agreement with the work of Khater [52], who suggested in his study that temperature, pH and EC of cow manure was 30˚C, 8.1, and 4.1 dSm$^{-1}$, and moisture content was 20%. According to Bhat et al. [54] and Lim et al. [55], EC values below 4.0 mS/cm are excellent for agricultural soil.

Our analysis of vermicompost through atomic absorption spectrophotometry indicated the increased level of microelements like sulfates, magnesium, manganese, copper, iron, and zinc compared to cow dung, which is available for plant growth and development, and outcomes are consistent with the outcomes of Jangra et al. [56]. The maximum values of calcium and

magnesium in described research were also consistent with the discoveries of Padmavathiamma et al. [57] and Jangra et al. [56]. C: N ratio is also a very important parameter for the maturity and stability of vermicompost. Our results agreed with the previous studies [58–60]. The loss of carbon and nitrogen ratio (11:1) is due to the mineralization process and consistent with the findings of Suthar and Singh [61] and Shak et al. [53]. They reported that carbon and nitrogen ratio less than 12 is desirable for agricultural purposes.

A recent study reported the presence of eleven plants growth-promoting vermibacteria (PGPVB) in the digestive tract of *E. fetida* which could be used as potential microbial biofertilizers to increase crop production [62, 63]. These vermibacteria possessed all plant growth-promoting traits and excrete in vermicast as well along with various substances which encourage plant growth and development. The current outcomes agreed with Yatoo et al. [64] and Adiloğlu et al. [65]. Vermicompost is utilized for the cultivation of various crops i.e. garlic, tomatoes, capsicum, peppermint, wheat, and maize [66–69]. Similar results were recorded in the current study that vermicompost was a perfect nutrient amendment to promote the growth of pansy. In this regard, our results demonstrated that all germination parameters of pansy i.e. early initiation of seed germination (3 days), minimum mean germination time (3.63±0.0124), maximum mean daily germination (0.32±0.020), highest speed of germination (9.63±0.081), the maximum percentage of germination (100±0.00), highest peak value (1.64±0.012) and maximum germination value (0.52±0.008) was recorded at 30% VC. In previous literature it was found that vermicompost prepared from lantana (*Lantana camara*), Salvinia (*Salvinia molesta*), ipomoea (*Ipomoea carnea)*, and parthenium (*Parthenium hysterophorus*) enhanced the germination of green gram, lady's finger, and cucumber when used in the concentration range of 25–35%. On the other hand, at higher levels of application it hindered the germination [70–73]. Based on our observations regarding seed germination, it is predictable that some bioactive compounds like phytohormones, humic acids, and other microbial metabolites exist in vermicompost and could be responsible for increased seed germination percentage, earlier emergence, and seedling growth. These outcomes are consistent with the previous literature [63, 65, 74, 75].

A significant effect of vermicompost on seedling parameters compared to control treatments were also recorded in the current study. We can say that 30% VC was a preferable dose for seedling growth parameters because the highest shoot length (3.66±0.47 cm, maximum number of leaves/plant (7.33±0.47), the maximum diameter of leaves (3.00±0.00 cm), the maximum leaves length (3.66±0.47 cm), maximum leaf area (6.33±0.47 cm), maximum seedling length (9.00±0.00 cm), and root length (4.66±0.47 cm) were recorded at this concentration. Previous literature suggested that seedling growth was enhanced in the VC amended media compared to control media and it was due to the presence of bioactive compounds in the VC that exerted the beneficial effects [63, 65, 74, 75]. Current research indicated the maximum vegetative and reproductive growth of pansy at 35% VC compared to controls (Sand + Soil + cow dung and Sand + Soil). As compared to other fertilizers, pansies cultivated in a slow-release fertilizer showed the most abundant flowering [76]. They found that pansy growth increased with an increasing dose of vermicompost, and our work also refers to the point that higher doses of slow-release fertilizer like vermicompost promotes overall growth pattern in plants. In the current research, the maximum vegetative growth parameters such as maximum plant height (12.33±0.47 cm), root volume (7.33±0.47 cm$^3$), number of leaves per plant (22.3 ±0.47), and diameter of leaves (4.16±0.23 cm), and flowering characteristics like maximum number of floral buds/plant (3.33 ±0.47), number of open flowers/plant (3.33±0.47), the diameter of flowers (7.83 ±0.23) were obtained at 35% VC. Similarly, at 35% VC initiation of floral buds after transplantation of plants occurs earlier at 74.6 days compared to control cow dung (81.4 days) and control soil (87.6 days). Our studies are thus parallel to the findings of Arancon

et al. [77] who studied the showed the significant effects of vermicompost on two ornamental plants, (marigolds, and petunias) Shafique et al. [18] also used vermicompost for marigold production in Azad Kashmir, Pakistan. However, our findings are contrary to the work of Lazcano and Dominguez [78], who reported that the highest dose of vermicompost decreased the survival rate of the pansy plants.

The current results illustrated that low nutrients were recorded in post-harvested germinated media compared to the media used before sowing and transplantation. The reduced nutrients indicate the uptake of nutrients by the plants from the germinating media is due to the more availability of NPK in vermicompost compared to cow dung. On the other hand, vermicompost containing germination media is more porous and soft in texture compared to control media. Therefore, vermicompost treatments showed significantly few nutrients in germination media after harvesting of pansy. Current findings reveal the maximum germination and vegetative growth of pansy at 30% VC and 35% VC due to maximum up taken nutrients by the plant's body, that's why 30% VC and 35% VC germinating media showed the least amount of nutrient profile compared to controls and other vermicompost containing germination media (Fig 3). Similarly, the nutrient analysis of both controls revealed the maximum nutrients retention in media due to compact and less porous texture, thus minimum nutrients are up taken by plants ensuing sluggish germination as well as vegetative growth.

## Conclusions

It was concluded that *E. fetida* is suitable earthworm species for the production of cow dung-based vermicompost. Produced vermicompost was rich in plant growth-promoting traits as well as nutrients along with considerable water holding capacity, EC, pH, temperature, and moisture contents which not only enhanced the seed germination, seedling and vegetative growth parameters but also improved the soil quality. The current research suggested that 30% VC is appropriate for the development of pansy nurseries whereas 35% VC is the best natural fertilizer for the maximum development of reproductive growth, vegetative growth, and flowering of pansy. Furthermore, vermicompost could be used as a potential substitute for a sustainable horticulture system.

## Supporting information

**S1 Graphical abstract.**
(JPG)

## Acknowledgments

Authors are thankful to the Soil testing laboratory of Soil Research institute, Gojra, AJ&K, Muzaffarabad, Pakistan for physicochemical analysis.

## Author Contributions

**Conceptualization:** Saiqa Andleeb.

**Data curation:** Irsa Shafique.

**Formal analysis:** Saiqa Andleeb, Farrukh Naeem, Shaukat Ali, Tariq Sultan.

**Funding acquisition:** Farrukh Naeem.

**Investigation:** Saiqa Andleeb, Mohammad Almas Abbasi.

**Methodology:** Shaukat Ali, Tauseef Tabassam, Tariq Sultan.

**Project administration:** Tariq Sultan.

**Supervision:** Saiqa Andleeb.

**Validation:** Irsa Shafique, Saiqa Andleeb, Tauseef Tabassam.

**Visualization:** Irsa Shafique, Saiqa Andleeb, Shaukat Ali, Tauseef Tabassam, Tariq Sultan, Mohammad Almas Abbasi.

**Writing – original draft:** Irsa Shafique.

**Writing – review & editing:** Saiqa Andleeb, Farrukh Naeem, Shaukat Ali, Tariq Sultan, Mohammad Almas Abbasi.

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
