## [Decision Letter · Decision Letter 0]

7 Jun 2022

PONE-D-22-12324Cow dung putrefaction via vermicomposting using Esinea fetida and its impact on seed germination and plant growth parameters of Viola wittrockiana (pansy)PLOS ONE

Dear Dr. Saiqa,

Thank you for submitting your manuscript to PLOS ONE. After careful consideration, we feel that it has merit but does not fully meet PLOS ONE’s publication criteria as it currently stands. Therefore, we invite you to submit a revised version of the manuscript that addresses the points raised during the review process.

ACADEMIC EDITOR: 

This manuscript discusses the impact of vermicompost on seed germination, seedlings, and vegetative growth parameters on plants. There are some serious concerns and points which must be improved. The manuscript fails to identify clearly major achievements in the topic in recent years, major research questions and future research needs. Unfortunately, the information provided in introduction and methodology is insufficient to properly interpret the study. In any case, the Discussion Section must be considerably improved, as it is too general. The manuscript also needs language editing, since it is difficult to understand what the authors are trying to convey in many places. I hope that my comments will help you to improve the quality of this study.

Title: Check the spelling of *Esinea fetida*. It should be *Eisenia fetida*

Abstract: Abstract should be rewritten by detailing the aim and concept of the paper. The abstract should state briefly the purpose of the research, the principal results and major conclusions.

Introduction: Very general and need to be elaborative to explore the actual philosophy to design the experiment. The introduction is insufficient to provide the state of the art in the topic. Hypothesis should be given. How this work is different from the available data?

The introduction part needs further improvement. The introduction of the paper must be extended and reformulated in order to provide a more comprehensive approach.

New references are missing in the introduction and discussion. Add 2018-2022 references accordingly

*Objectives of the present study is poorly written. In the last paragraph of the introduction, write the specific objectives of this work clearly, as points (i)..... (ii)....(iii) ......and (iv).

Results and Discussion:

The manuscript does not provide interesting and technically sound discussion; it would be better to use more recent references in discussion.  Authors are suggested to add discussion by explaining trends in the obtained results along with the possible mechanisms behind the trends.

Authors are suggested to draw major inferences/primary conclusions followed by the secondary conclusions/ recommendations reached through the critical analysis/ investigation of the study.

I recommend you to revise the manuscript after considering these points along with the reviewer’s comments.Please ensure that your decision is justified on PLOS ONE’s publication criteria and not, for example, on novelty or perceived impact.

We look forward to receiving your revised manuscript.

Kind regards,

Sartaj Ahmad Bhat, Ph.D

Academic Editor

PLOS ONE

Journal Requirements:

2. Thank you for submitting the above manuscript to PLOS ONE. During our internal evaluation of the manuscript, we found significant text overlap between your submission and the following previously published works, some of which you are an author.

- http://jdguez.webs.uvigo.es/wp-content/uploads/2012/01/the-use-of-vermicompost.pdf

- https://revistas.inia.es/index.php/sjar/article/view/1412

- https://jast-journal.springeropen.com/articles/10.1186/s40543-017-0112-2?optIn=false

- https://www.blackdiamondvermicompost.com/wp-content/uploads/2010/12/EFFECTS-OF-VERMICOMPOSTS-ON-PLANT-GROWTH.pdf

- https://en.wikipedia.org/wiki/Pansy

- https://moam.info/viola-wittrockiana-gams_5c64e2a7097c471b788b459f.html

- https://pubmed.ncbi.nlm.nih.gov/33490670/

- http://ijsrm.humanjournals.com/wp-content/uploads/2018/03/8.Sumathi-S-Pawlin-Vasanthi-Joseph.pdf

- https://ijcmas.com/7-10-2018/Anil%20Kumar,%20et%20al.pdf

- https://www.mdpi.com/2071-1050/10/4/1205/html

-http://hortsci.ashspublications.org/content/47/12/1722.full

Please revise the manuscript to rephrase the duplicated text, cite your sources, and provide details as to how the current manuscript advances on previous work. Please note that further consideration is dependent on the submission of a manuscript that addresses these concerns about the overlap in text with published work.

Reviewers' comments:

Reviewer's Responses to Questions

**Comments to the Author**

1. Is the manuscript technically sound, and do the data support the conclusions?

Reviewer #1: Yes

Reviewer #2: Yes

2. Has the statistical analysis been performed appropriately and rigorously? 

Reviewer #1: Yes

Reviewer #2: Yes

3. Have the authors made all data underlying the findings in their manuscript fully available?

Reviewer #1: Yes

Reviewer #2: Yes

4. Is the manuscript presented in an intelligible fashion and written in standard English?

Reviewer #1: Yes

Reviewer #2: Yes

5. Review Comments to the Author

Reviewer #1: Dear Authors,

Your manuscript need significant improvement particularly in discussion section to be more compact and reduce the length. Simply delete general or repeated or not necessary sentences and texts.

Reviewer #2: The manuscript title "Cow dung putrefaction via vermicomposting using Esinea fetida and its impact on seed germination and plant growth parameters of Viola wittrockiana (pansy)" was written very well. However, authors need to address to the review comments for improving the quality of the manuscript prior to the publication process.

1. Recent references should be incorporated in both introduction and discussion part

2. Biomass and reproduction data of earthworm is missing.

3. Reproduction and growth rate should be presented in tabular or graphical form

4. Grammar and spelling mistakes should be considered/removed in revised manuscript

5. conclusion could be improved

6. Table 1: Need to mention unit of EC

7. Table 2: .Need to mention the unit of germination speed

8.Please mention the representation of “bbb” in Statistical icons

6. PLOS authors have the option to publish the peer review history of their article (what does this mean?). If published, this will include your full peer review and any attached files.

Reviewer #1: No

Reviewer #2: No

---

## [Author Response · Author response to Decision Letter 0]

14 Jul 2022

ACADEMIC EDITOR: 

This manuscript discusses the impact of vermicompost on seed germination, seedlings, and vegetative growth parameters on plants. There are some serious concerns and points which must be improved. The manuscript fails to identify clearly major achievements in the topic in recent years, major research questions and future research needs. Unfortunately, the information provided in introduction and methodology is insufficient to properly interpret the study. In any case, the Discussion Section must be considerably improved, as it is too general. The manuscript also needs language editing, since it is difficult to understand what the authors are trying to convey in many places. I hope that my comments will help you to improve the quality of this study.

Title: Check the spelling of Esinea fetida. It should be Eisenia fetida

Has been corrected

Abstract: Abstract should be rewritten by detailing the aim and concept of the paper. The abstract should state briefly the purpose of the research, the principal results and major conclusions.

Has been modified

Introduction: Very general and need to be elaborative to explore the actual philosophy to design the experiment. The introduction is insufficient to provide the state of the art in the topic. Hypothesis should be given. How this work is different from the available data?

The introduction part needs further improvement. The introduction of the paper must be extended and reformulated in order to provide a more comprehensive approach.

New references are missing in the introduction and discussion. Add 2018-2022 references accordingly

*Objectives of the present study is poorly written. In the last paragraph of the introduction, write the specific objectives of this work clearly, as points (i)..... (ii)....(iii) ......and (iv).

Introduction has been modified along with recent references. 

Results and Discussion:

The manuscript does not provide interesting and technically sound discussion; it would be better to use more recent references in discussion. Authors are suggested to add discussion by explaining trends in the obtained results along with the possible mechanisms behind the trends.

Authors are suggested to draw major inferences/primary conclusions followed by the secondary conclusions/ recommendations reached through the critical analysis/ investigation of the study.

Manuscript has been modified along with recent references.

5. Review Comments to the Author

Reviewer #1: Dear Authors,

Your manuscript need significant improvement particularly in discussion section to be more compact and reduce the length. Simply delete general or repeated or not necessary sentences and texts.

Thanks for the suggestions to improve the quality of manuscript. All data have been improved in revised manuscript. Repeated data has been removed.

Reviewer #2: The manuscript title "Cow dung putrefaction via vermicomposting using Esinea fetida and its impact on seed germination and plant growth parameters of Viola wittrockiana (pansy)" was written very well. However, authors need to address to the review comments for improving the quality of the manuscript prior to the publication process.

Thanks for the positive recommendations. 

1. Recent references should be incorporated in both introduction and discussion part

Has been incorporated

2. Biomass and reproduction data of earthworm is missing.

Has been incorporated

3. Reproduction and growth rate should be presented in tabular or graphical form

Has been presented as figure 1

4. Grammar and spelling mistakes should be considered/removed in revised manuscript

Has been removed

5. conclusion could be improved

Has been improved

6. Table 1: Need to mention unit of EC

Has been mentioned

7. Table 2: Need to mention the unit of germination speed

Has been mentioned

8.Please mention the representation of “bbb” in Statistical icons

Has been mentioned in statistical analysis section

---

## [Decision Letter · Decision Letter 1]

22 Sep 2022

PONE-D-22-12324R1Cow dung putrefaction via vermicomposting using Eisenia fetida and its influence on seed sprouting and vegetative growth of Viola wittrockiana (pansy)PLOS ONE

Dear Dr. Saiqa,

Thank you for submitting your manuscript to PLOS ONE. After careful consideration, we feel that it has merit but does not fully meet PLOS ONE’s publication criteria as it currently stands. Therefore, we invite you to submit a revised version of the manuscript that addresses the points raised during the review process.

ACADEMIC EDITOR: The manuscript still needs some revisions. There are several uses of unclear language, highly general statements, and allusions to trends in the data without specific support. 

We look forward to receiving your revised manuscript.

Kind regards,

Sartaj Ahmad Bhat, Ph.D

Academic Editor

PLOS ONE

Journal Requirements:

Reviewers' comments:

Reviewer's Responses to Questions

**Comments to the Author**

1. If the authors have adequately addressed your comments raised in a previous round of review and you feel that this manuscript is now acceptable for publication, you may indicate that here to bypass the “Comments to the Author” section, enter your conflict of interest statement in the “Confidential to Editor” section, and submit your "Accept" recommendation.

Reviewer #3: All comments have been addressed

Reviewer #4: (No Response)

2. Is the manuscript technically sound, and do the data support the conclusions?

Reviewer #3: Yes

Reviewer #4: Partly

3. Has the statistical analysis been performed appropriately and rigorously? 

Reviewer #3: Yes

Reviewer #4: No

4. Have the authors made all data underlying the findings in their manuscript fully available?

Reviewer #3: Yes

Reviewer #4: No

5. Is the manuscript presented in an intelligible fashion and written in standard English?

Reviewer #3: Yes

Reviewer #4: Yes

6. Review Comments to the Author

Reviewer #3: The authors have addressed the comments and the manuscript has been improved. One minor comment: Please don't use double sign for showing statistical significance of the data. Simply '*' or '***' is enough and generally used in literature. There is no need for other symbols.

Reviewer #4: The authors have addressed the review suggestions to a certain extent; yet the effort is inadequate. Moreover, cow dung itself is a source of bioavailable nutrients because it decomposes quickly. Please porvide scientific explanation, why putrefaction of cow dung is nececcasry. Also, the introduction must discuss the recent knowledge on vermicomposting. Some papers which are published in recent years should be dicussed and cited, for example,

"Epigenetic regulations enhance adaptability and valorization efficiency in Eisenia fetida and Eudrilus eugeniae during vermicomposting of textile sludge: Insights on repair mechanisms of metal-induced genetic damage and oxidative stress. Bioresource Technology (Elsevier), 2022, 345: 126493"

"Appraisal of lignocellusoic biomass degrading potential of three earthworm species using vermireactor mediated with spent mushroom substrate: Compost quality, crystallinity, and microbial community structural analysis. Science of the Total Environment, (2019), doi: " ext-link-type="uri" xlink:type="simple">https://doi.org/10.1016/j.scitotenv.2019.135215"

"Application of drum compost and vermicompost to improve soil health, growth, and yield parameters for tomato and cabbage plants. Journal of Environmental Management (Elsevier), (2017), 200: 243-252"

"Vermicompost and farm yard manure improves food quality, anti-oxidant and antibacterial potential of Cajanus cajan (L, Mill sp.) leaves. Journal of the Science of Food and Agriculture (Wiley) (2016) 97: 956–966"

7. PLOS authors have the option to publish the peer review history of their article (what does this mean?). If published, this will include your full peer review and any attached files.

Reviewer #3: No

Reviewer #4: No

---

## [Author Response · Author response to Decision Letter 1]

27 Sep 2022

Response to Reviewers

Reviewer #3: The authors have addressed the comments and the manuscript has been improved. One minor comment: Please don't use double sign for showing statistical significance of the data. Simply '*' or '***' is enough and generally used in literature. There is no need for other symbols.

Thanks for the endorsement and recommendation. In the current study authors used two different symbols to represent the differences among the treatments and control groups i.e. ‘*’ represents a significant change among Tcc and VC-treated groups and ‘¥’ represents a significant change among To and VC-treated groups. Otherwise we don’t need to use these symbols.

Reviewer #4: The authors have addressed the review suggestions to a certain extent; yet the effort is inadequate. Moreover, cow dung itself is a source of bioavailable nutrients because it decomposes quickly. Please provide scientific explanation, why putrefaction of cow dung is nececcasry. 

Thanks for the comments. You are absolutely right that cow dung is itself is a source of bioavailable nutrients but its decomposition is necessary due to following reasons. Manure of cow/horse/chicken is not only too much hot but also contain too high nitrogen contents which is not possible for plants to tolerate. Both temperature and excess nitrogen burns the plants. Secondly, Excess of methane generation and release of toxic gases pollute the environment. 

Also, the introduction must discuss the recent knowledge on vermicomposting. Some papers which are published in recent years should be dicussed and cited, for example,

"Epigenetic regulations enhance adaptability and valorization efficiency in Eisenia fetida and Eudrilus eugeniae during vermicomposting of textile sludge: Insights on repair mechanisms of metal-induced genetic damage and oxidative stress. Bioresource Technology (Elsevier), 2022, 345: 126493"

"Appraisal of lignocellusoic biomass degrading potential of three earthworm species using vermireactor mediated with spent mushroom substrate: Compost quality, crystallinity, and microbial community structural analysis. Science of the Total Environment, (2019), doi: https://doi.org/10.1016/j.scitotenv.2019.135215"

"Application of drum compost and vermicompost to improve soil health, growth, and yield parameters for tomato and cabbage plants. Journal of Environmental Management (Elsevier), (2017), 200: 243-252"

"Vermicompost and farm yard manure improves food quality, anti-oxidant and antibacterial potential of Cajanus cajan (L, Mill sp.) leaves. Journal of the Science of Food and Agriculture (Wiley) (2016) 97: 956–966"

Thanks for the suggestions. All references have been incorporated in revised manuscript.

---

## [Decision Letter · Decision Letter 2]

15 Dec 2022

Cow dung putrefaction via vermicomposting using Eisenia fetida and its influence on seed sprouting and vegetative growth of Viola wittrockiana (pansy)

PONE-D-22-12324R2

Dear Dr. Saiqa,

We’re pleased to inform you that your manuscript has been judged scientifically suitable for publication and will be formally accepted for publication once it meets all outstanding technical requirements.

Kind regards,

Sartaj Ahmad Bhat, Ph.D

Academic Editor

PLOS ONE

Additional Editor Comments (optional):

Reviewers' comments:

Reviewer's Responses to Questions

**Comments to the Author**

1. If the authors have adequately addressed your comments raised in a previous round of review and you feel that this manuscript is now acceptable for publication, you may indicate that here to bypass the “Comments to the Author” section, enter your conflict of interest statement in the “Confidential to Editor” section, and submit your "Accept" recommendation.

Reviewer #4: (No Response)

2. Is the manuscript technically sound, and do the data support the conclusions?

Reviewer #4: Yes

3. Has the statistical analysis been performed appropriately and rigorously? 

Reviewer #4: Yes

4. Have the authors made all data underlying the findings in their manuscript fully available?

Reviewer #4: Yes

5. Is the manuscript presented in an intelligible fashion and written in standard English?

Reviewer #4: Yes

6. Review Comments to the Author

Reviewer #4: The paper has been adequately improved by addressing all the comments. I congratulate the authors for their effort.

7. PLOS authors have the option to publish the peer review history of their article (what does this mean?). If published, this will include your full peer review and any attached files.

Reviewer #4: **Yes: **Satya Sundar Bhattacharya

---

## [Editor Report · Acceptance letter]

6 Jan 2023

PONE-D-22-12324R2 

Cow dung putrefaction via vermicomposting using *Eisenia fetida* and its influence on seed sprouting and vegetative growth of *Viola wittrockiana* (pansy) 

Dear Dr. Andleeb:

I'm pleased to inform you that your manuscript has been deemed suitable for publication in PLOS ONE. Congratulations! Your manuscript is now with our production department. 

Kind regards, 

on behalf of

Dr. Sartaj Ahmad Bhat 

Academic Editor

PLOS ONE